# Parametric CAD modeling for open source scientific hardware: Comparing OpenSCAD and FreeCAD Python scripts

**Felipe Machado** **\*, Norberto Malpica**, **Susana Borromeo**

Area of Electronics Technology, Universidad Rey Juan Carlos, Móstoles, Spain

☯ These authors contributed equally to this work.
\* felipe.machado@urjc.es

## Abstract

Open source hardware for scientific equipment needs to provide source files and enough documentation to allow the study, replication and modification of the design. In addition, parametric modeling is encouraged in order to facilitate customization for other experiments. Parametric design using a solid modeling programming language allows customization and provides a source file for the design. OpenSCAD is the most widely used scripting tool for parametric modeling of open source labware. However, OpenSCAD lacks the ability to export to standard parametric formats; thus, the parametric dimensional information of the model is lost. This is an important deficiency because it is key to share the design in the most accessible formats with no information loss. In this work we analyze OpenSCAD and compare it with FreeCAD Python scripts. We have created a parametric open source hardware design to compare these tools. Our findings show that although Python for FreeCAD is more arduous to learn, its advantages counterbalance the initial difficulties. The main benefits are being able to export to standard parametric models; using Python language with its libraries; and the ability to use and integrate the models in its graphical interface. Thus, making it more appropriate to design open source hardware for scientific equipment.

**Data Availability Statement:** Source files and CAD files are available in these two repositories: For

## Introduction

Over the last years there have been a movement towards creating and sharing Open Source Hardware (OSH). This trend has been empowered by open source hardware projects such as the RepRap 3D printers [1] and the Arduino platform [2], which have made manufacturing and electronic technology accessible and affordable. As a result to this movement many efforts have been made to define open source hardware and set its best practices [3] [4] [5].

Inspired by this movement, an engineering research area has emerged to develop open source scientific hardware and laboratory equipment [6] [7] [8] [9] [10]. Open source scientific hardware not only allows a more affordable laboratory equipment, but also contributes to the development of Open Science by facilitating the replication and comparison of the scientific

OpenSCAD: https://github.com/felipe-m/oscad_filter_stage For FreeCAD: https://github.com/felipe-m/freecad_filter_stage.

**Funding:** N.M. and S.B. were supported by Spanish Department of Economy and Competitiveness grant RTC-2015–4167-1 (http://www.mineco.gob.es/; this department changed its name); and Spanish Department of Science, Innovation and Universities grant RTC-2017-6218-1 (http://www.ciencia.gob.es/). The funders had no role in study design, data collection and analysis, decision to publish, or preparation of the manuscript.

**Competing interests:** The authors have declared that no competing interests exist.

experiments. Moreover, it favors the enhancement of experiments by letting others to improve and customize the devices for different purposes.

As stated in the Open Source Hardware Statement of Principles [3], open source hardware is hardware for which the design is made publicly available so that anyone can study, modify, distribute, make, and sell the design or hardware based on that design.

Making OSH is not a matter of just providing an open source license for the hardware; in addition, design files, documentation and any source code should be available in the preferred format for making modifications to them and with an acceptable open license. Furthermore, it is encouraged that these files are made editable with free and open source software (FOSS) [4] [7] [11] [12].

Ideally, in order to maximize the ability of individuals to use and make the hardware, open source hardware should provide unrestricted content, and use readily-available components and materials, standard processes, open infrastructure and open-source design tools [3] [7].

The Open Source Hardware Definition [3] is based on the Open Source Definition for Open Source software (OSS) [13] and adapted to the realms of tangible things. However, there are some aspects of hardware that differ from software. First, unlike software, most of a hardware project will fall within the scope of patent law rather than copyright law [14]. Secondly, hardware designs demand a wider range of expertise because new areas of knowledge are involved, particularly when including mechanics, electronics and software [15] [16].

And lastly, the source in hardware is not as clearly defined as in software, as Bonvoisin et al. show from their study of several OSH products [5]. Their analysis reveals a wide range of interpretations of open source hardware, exposing that many projects lack enough documentation to replicate or modify the product.

This last issue unveils one of the main problems for OSH: the deficiency in documentation to consider it truly open [5] [15] [17]. Bonvoisin et al. [5] assert that while software source code unambiguously defines the software, i.e. the product; the source for tangible things is not so clearly defined. Consequently, unlike OSH, the openness of OSS is implicitly fulfilled just by providing the source code. Although this statement can be considered accurate; it could nevertheless be argued that proper documentation for OSS is also a need, especially for large projects [18] [19]. It is unpractical to try to understand large and complex OSS projects without adequate documentation. The 2017 Open Source GitHub Survey supports this idea by highlighting that incomplete documentation is the biggest concern for OSS [20].

In addition to the stated OSH requirements, Oberloier et al. [7] encourage parametric design of OSH for scientific equipment. Parametric design enables customization by providing the flexibility to alter the model dimensions for other experimental purposes. Parametric models can be created using script-based computer-aided design (CAD) tools such as OpenSCAD [21]. OpenSCAD is an open source software tool that may be considered the defacto standard for OSH parametric design [8] as it has been widely used to create parametric OSH for laboratory equipment [7] [8] [22] [23] [24] [25] [26] [27].

The benefits of using a script-based CAD tool such as OpenSCAD are twofold. First, it allows model customization by means of a parametric design. Secondly, it provides a source code for the mechanical design, making it more similar to software. Thereby, addressing the hardware absence of source code that Bonvoisin et al. pointed out [5]. Having a source code for the hardware may mitigate the difficulties of OSH projects in defining their true openness. Besides, it allows OSH projects to use software management tools such as control version, software documentation and collaborative tools. In a way, the mechanical design would be similar to the electronic design of digital circuits with hardware description languages; although in the electronic field there is less availability of FOSS design tools to cover the whole process [16] [28]

Unfortunately, one of the major limitations of OpenSCAD is its inability to export to industry standard CAD file formats such as STEP [29] [30]. Although OpenSCAD can export the models to tessellated formats that can be read in most CAD tools, these formats are only approximate because they have lost their parametric dimensions. As many authors suggest [12] [16] [31] [32], providing a standard file format is crucial since it allows others to modify the OSH design with different CAD tools. This is a critical issue since the user may not be familiar with a script-based CAD tool such as OpenSCAD.

In this paper we review script-based FOSS CAD tools in order to find an alternative to OpenSCAD. Among these tools, we have found FreeCAD [33] to be a suitable candidate, since it is able to export to standard parametric CAD formats. Although FreeCAD is mainly used through its graphical user interface (GUI), it also allows creating CAD models using Python programming language [34]. Therefore, in this paper we analyze OpenSCAD, which is the most extensively used tool for modeling open scientific equipment with a programming language, and compare it with Python scripts for FreeCAD (hereafter FreeCAD Python).

In order to compare these tools, we have created a configurable OSH model of a motorized optical filter stage. The filter stage has four components that have been modeled with both OpenSCAD and FreeCAD Python. Modeling these parts with both tools has allowed us to analyze their strengths and weakness.

As a result of our analysis we suggest that although FreeCAD Python has a larger learning curve, it has an extensive set of features that makes it more suitable and powerful for modeling open source labware.

This paper is organized as follows. In the next section, script-based FOSS CAD tools are reviewed. Next, the CAD models used as a test-bench are presented. Afterwards, the parametrization of the CAD models is described. The following section discusses the benefits and drawbacks of the two CAD tools analyzed: OpenSCAD and FreeCAD Python. Conclusions are drawn in the final section.

## Script-based CAD tools for parametric modeling

In this study we analyze CAD tools for open source scientific equipment. Computer software for solid modeling can be broadly classified into two types: parametric and free form mesh modelers. The former create exact and complex mathematical data structures and models. The latter generate a simple mesh of polygonal surfaces, also known as tessellated geometries. Having the purpose of designing mechanical pieces, whose dimensions must be accurate, parametric modelers are better fitted for the task. In addition to using a parametric modeler, parametric modeling is also recommended for OSH labware [7]. In parametric modeling, designers define the size, shape and position of geometric features and assembly components in term of parameters [35] [36] [37] [38]. Since scripting is particularly appropriate for parametric design [31] [39] [40], we have limited the analysis to CAD tools with scripting capabilities. Finally, following the best practices for OSH [4] [7] [11] [12] we have only selected free and open source software.

We had initially chosen OpenSCAD for being the most widely CAD tool used in parametric modeling for open source scientific equipment. However, we needed to find an alternative because OpenSCAD is not able to export to a standard parametric CAD file format such as STEP [30].

There are other available FOSS CAD tools for solid modeling using a programming language. Examples of these tools are BRL-CAD [41], CadQuery [42], PythonOCC [43], FreeCAD [33], ImplicitCad [44], OpenJSCAD [45] and Blender [46].

Blender was rejected for being a free form mesh modeler, which is more suitable to create natural or artistic designs rather than precise mechanical designs.

Both OpenJSCAD and ImplicitCad are similar to OpenSCAD. OpenJSCAD uses JavaScript as a programming language. ImplicitCad is an entirely separate project from OpenSCAD, but it has an OpenSCAD language interpreter. Thus, OpenSCAD designs can be imported to ImplicitCad. However, neither OpenJSCAD nor ImplicitCad are able to export to standard parametric file formats. Therefore, they have not been considered in the analysis as they do not provide a solution for the main OpenSCAD disadvantage.

FreeCAD is a very active project that is able to export to standard parametric CAD formats. Besides, it allows both GUI modeling and script-based modeling using Python. FreeCAD can be totally controlled by Python scripts and provides an Application Programming Interface (API) for solid modeling using Python scripts. Therefore, FreeCAD can be used by both graphical designers and CAD programmers.

PythonOCC is a Python library that provides 3D modeling features. PythonOCC is a wrapper of the OCCT library [47], which is the same geometric modeling kernel that FreeCAD uses. PythonOCC is able to export to standard parametric formats; nonetheless, as a Python library without graphical interface, it may be too challenging for designers with less programming experience.

BRL-CAD is a powerful solid modeler that has been active for more than 30 years; however, it is an expert oriented tool with a long learning curve. Since the OSH labware designer is not necessarily an expert CAD designer, we consider that an easier tool would be preferable.

CadQuery is a Python library that allows creating parametric models with a reduced amount of code. CadQuery has two working versions: v1.2 [42] and v2.0 [48]. The former is built on top of FreeCAD API and can be installed as a FreeCAD workbench. The latter is built on PythonOCC. Both versions are able to export to standard parametric formats.

CadQuery v1.2 can be easily integrated into FreeCAD and be used with FreeCAD graphical interface. Thus, CadQuery v1.2 will be included in the analysis as a part of FreeCAD. On the other hand, CadQuery v2.0 is a new fork of CadQuery that is independent from FreeCAD. We have not included it in our analysis because it has been recently released (December 2018) and it is in an early development stage; however, we consider that it could be a promising alternative. At the present day the two versions remain compatible; hence, the CadQuery filter stage model and the code snippets shown in this paper are valid for both versions.

Table 1 shows a summary of the FOSS CAD tools with scripting capabilities. For each tool, the central column shows if it is able to export to the STEP standard parametric format. In addition, the last column shows an important characteristic of the tool.

As it has been said, OpenSCAD is the prevalent CAD tool for OSH scientific equipment; however, OpenSCAD has a major drawback related to exporting to standard exchange formats. In order to find an alternative CAD tool, this article compares OpenSCAD with FreeCAD Python. In this analysis, we will also include CadQuery v1.2 as a part of FreeCAD workbenches.

## Open hardware models created as test-bench

In order to compare the proposed CAD scripting tools, we have designed a parametric OSH motorized optical filter stage. The stage allows positioning an optical filter along a linear axis. The stage has four printable pieces modeled in both OpenSCAD and FreeCAD Python. They have also been modeled using CadQuery library v1.2, which is installed as a workbench in FreeCAD. The source code of the models is available in two software repositories: one for the OpenSCAD models [49] and the other for the FreeCAD and CadQuery models [50].

**Table 1. FOSS CAD tools with scripting capabilities.**

| CAD tool | STEP export | Main characteristic |
| --- | --- | --- |
| OpenSCAD | No | Widely used for parametric OSH labware |
| Blender | No | Not intended for mechanical CAD |
| OpenJSCAD | No | JavaScript, similar to OpenSCAD |
| ImplictiCad | No | Similar to OpenSCAD |
| FreeCAD | Yes | Both GUI and scripted modeling |
| PythonOCC | Yes | Just for scripting, no GUI |
| BRL-CAD | Yes | Expert oriented |
| CadQuery v1.2 | Yes | FreeCAD workbench |
| CadQuery v2.0 | Yes | Recently released, built on PythonOCC |

Positive features for designing OSH labware are shaded in light blue. Negative features are shaded in light red. Non shaded cell describe neutral features or that could be considered positive or negative.

These pieces are: a stepper motor bracket; an optical filter holder that can be attached to a linear guide; and two pieces that are parts of a timing belt tensioner. The models are configurable; hence, they can be easily modified by just changing the model parameters. Fig 1 shows an arrangement example of the configurable filter stage with its printable parts highlighted.

Following, the printable pieces used for the analysis will be explained with more detail. For more information about the designs, the software repositories contain step-by-step tutorials explaining the generation of the models [49] [50].

## Motor bracket

The purpose of this piece is mounting the stepper motor to a structure. It has four holes aligned to the mounting holes of the motor, and it has two slots to provide flexibility in securing the bracket to a surface. Besides, the bracket is reinforced along the sides to be able to hold heavy motors (Fig 2A). Fig 2B illustrates how the motor is mounted on an aluminum profile.

## Filter holder

The filter holder is a single printable piece to place an optical filter. It has various holes to attach the piece to a linear guide. It has more bolt holes than needed to provide flexibility to

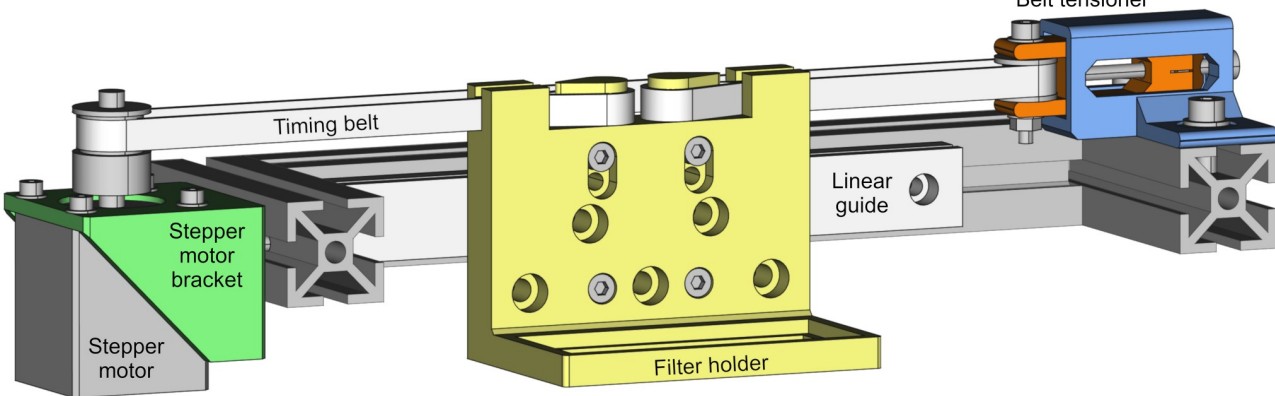

**Fig 1. Arrangement example of the configurable filter stage.** Printable parts are highlighted with colors: motor bracket (green), filter holder (yellow), belt tensioner (blue and orange).

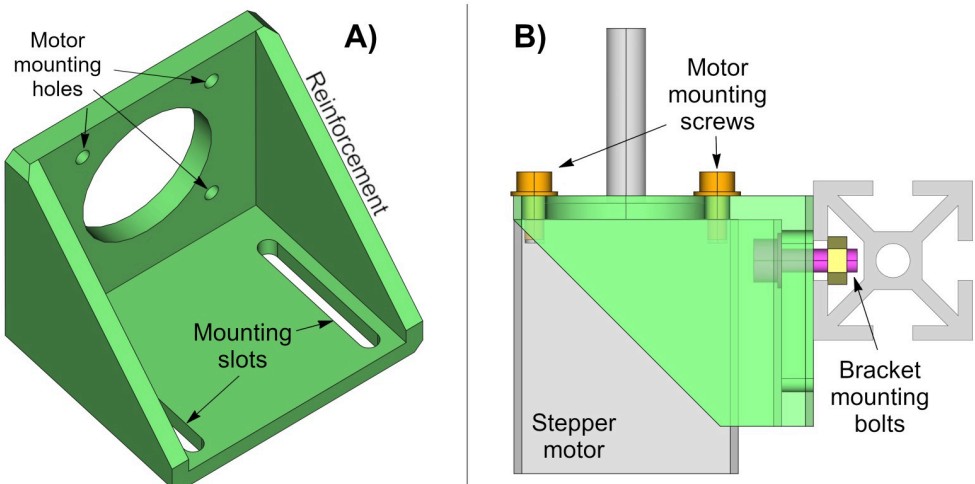

**Fig 2. Motor bracket.** (A) Drawing of the motor bracket. (B) Side view of the motor mount. The bracket and the stepper motor are partially transparent to show the internal parts.

use it with different linear guides or other structures. The piece has two timing belt clamps to pull the filter along the desired direction (Fig 3A). Fig 3B shows the exploded view of the filter holder mounted on a linear guide.

## Belt tensioner

The belt tensioner is a more complex unit because it is composed by two printable parts and some other elements such as an idler pulley, and a few bolts, nuts and washers. The printable parts are an idler pulley tensioner and a tensioner holder. Fig 4 shows the belt tensioner with its parts from two different perspectives. The tensioner holder (blue) and the idler tensioner (orange) are the printable parts.

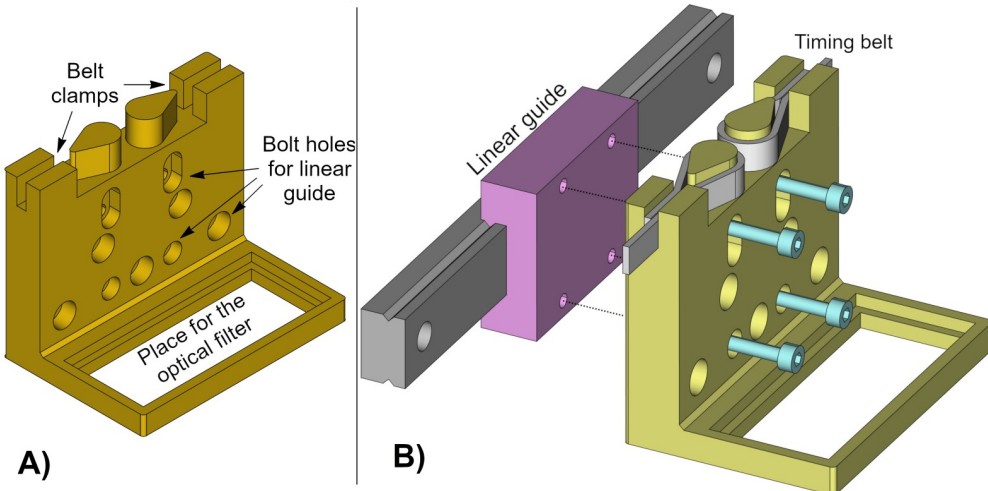

**Fig 3. Filter holder.** (A) Drawing showing the filter holder parts. (B) Exploded view of the filter holder assembly to a linear guide.

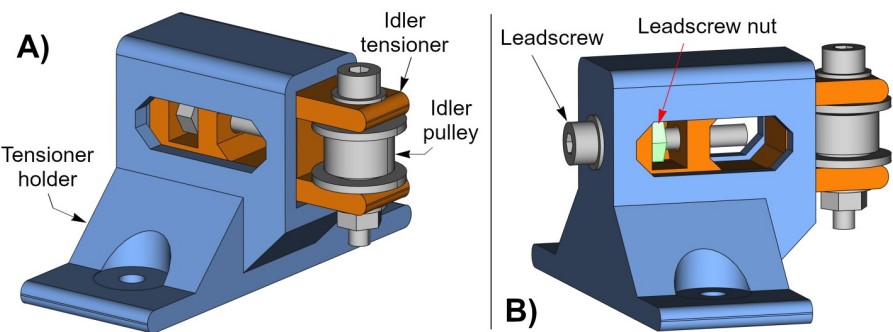

**Fig 4. Belt tensioner.** (A) Belt tensioner side-front view. (B) Belt tensioner side-back view.

To help understanding of the belt tensioner assembly, Fig 5 shows an exploded view of its parts.

The belt tensioner works by turning the leadscrew. The leadscrew nut cannot rotate because it is inserted inside the idler tensioner; thus, depending on the direction of the leadscrew rotation, the belt tensioner will retract (Fig 6A) or extend (Fig 6B) the idler tensioner (orange). As a consequence, this operation tightens or loosens the timing belt.

## 3D printing the models

These four pieces have been designed to be 3D printed without support structures. Printing without support is faster, minimizes waste material, produces better surface finishing, reduces

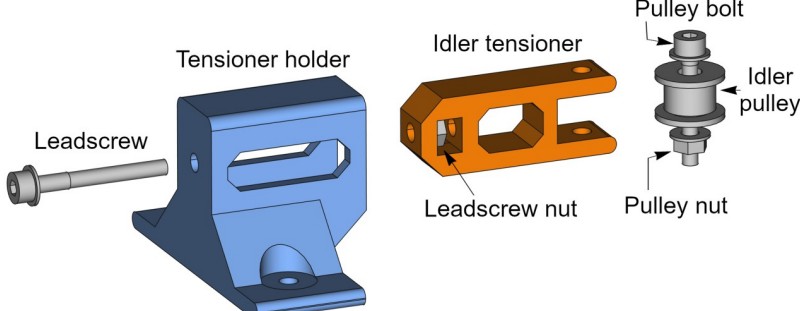

**Fig 5. Exploded view of the belt tensioner.** The two printable parts are drawn in colors: tensioner holder (blue) and idler tensioner (orange).

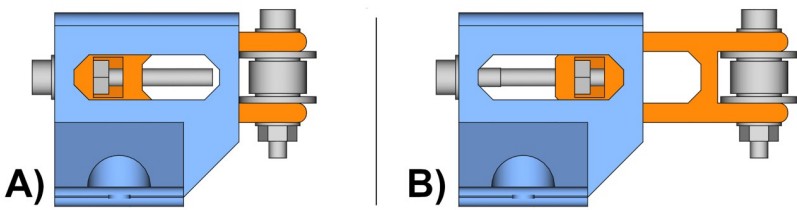

**Fig 6. Belt tensioner operation.** The leadscrew tightens or loosens the idler tensioner (orange). (A) Side view of the retracted belt tensioner. (B) Side view of the extended belt tensioner.

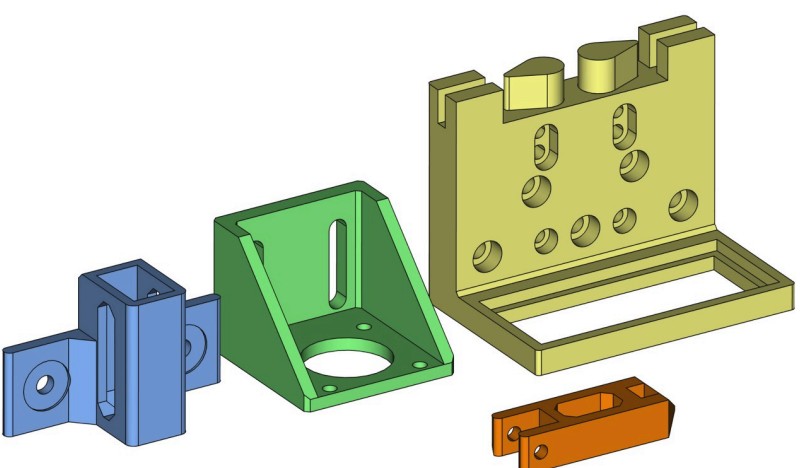

**Fig 7. 3D printing orientation of the four printable models to avoid supports.**

post processing work and thus, decreases the probability of damaging the piece due to the post processing.

Fig 7 shows the orientation to print the pieces without support structures.

## Parametric design

One of the main advantage of script-based modeling is the ability to change parameters values and generate variations of the original model with little effort. Programmed modeling can be time consuming compared to traditional graphical design, but the effort pays off when many variations of the model are needed, or when not all the specifications have been set from the beginning; thus, the final dimensions may change.

We have defined several parameters for the proposed models. Following, the main parameters for the motor bracket, the filter holder and the belt tensioner will be described.

### Motor bracket parameters

Depending on the stepper motor size the resulting bracket will have different dimensions; hence, the main parameter is the standardized NEMA size of the motor. For example, the NEMA size will define the parameter `motorbolt_sep` shown in Fig 8 and will also determine the minimum inner space for the motor. Other parameters such as walls thickness, the length and width of the slots can be modified. Some of the main parameters are shown in Fig 8.

As an example, Fig 9 shows the resulting brackets for two different motor sizes. Note that the length of the slots have also been modified.

### Filter holder parameters

The filter holder can be configured to carry different filter sizes and to be attached to almost any bolt arrangement. For this purpose, the bolt hole positions and sizes are configurable. Fig 10 shows the main filter holder parameters. There are other parameters that are less critical and are documented in the source code.

As an example, Fig 11 shows two filter holders that have been generated with different parameters.

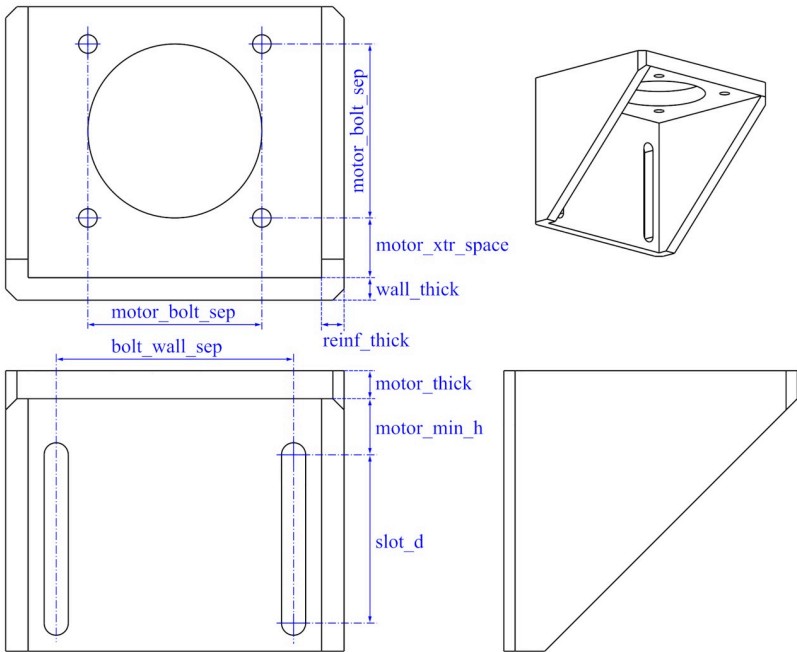

**Fig 8. Motor bracket main parameters.**

## Belt tensioner parameters

Unlike the motor bracket and the filter holder, the belt tensioner is composed by various elements; consequently, some of the dimensions of these elements are interdependent. In these cases, parametric design plays a significant role because it allows to establish the dependencies

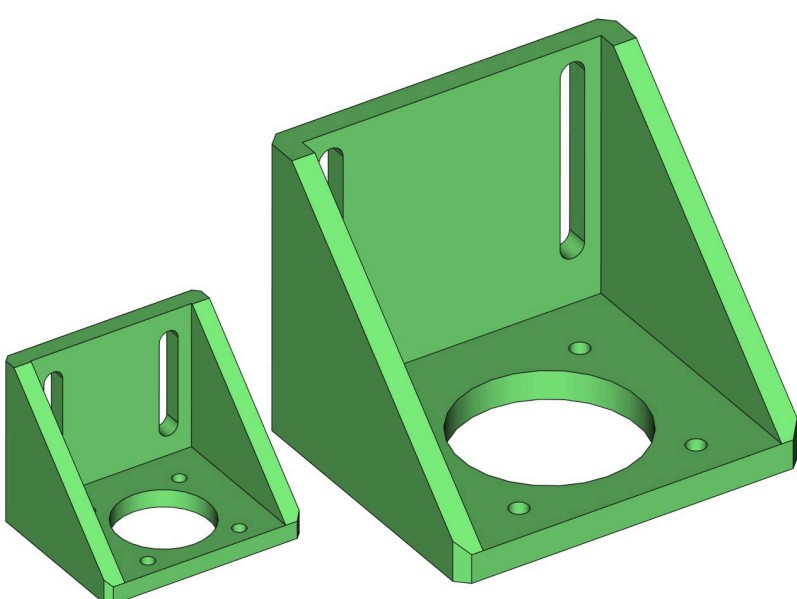

**Fig 9. Two brackets for different stepper motor sizes.**

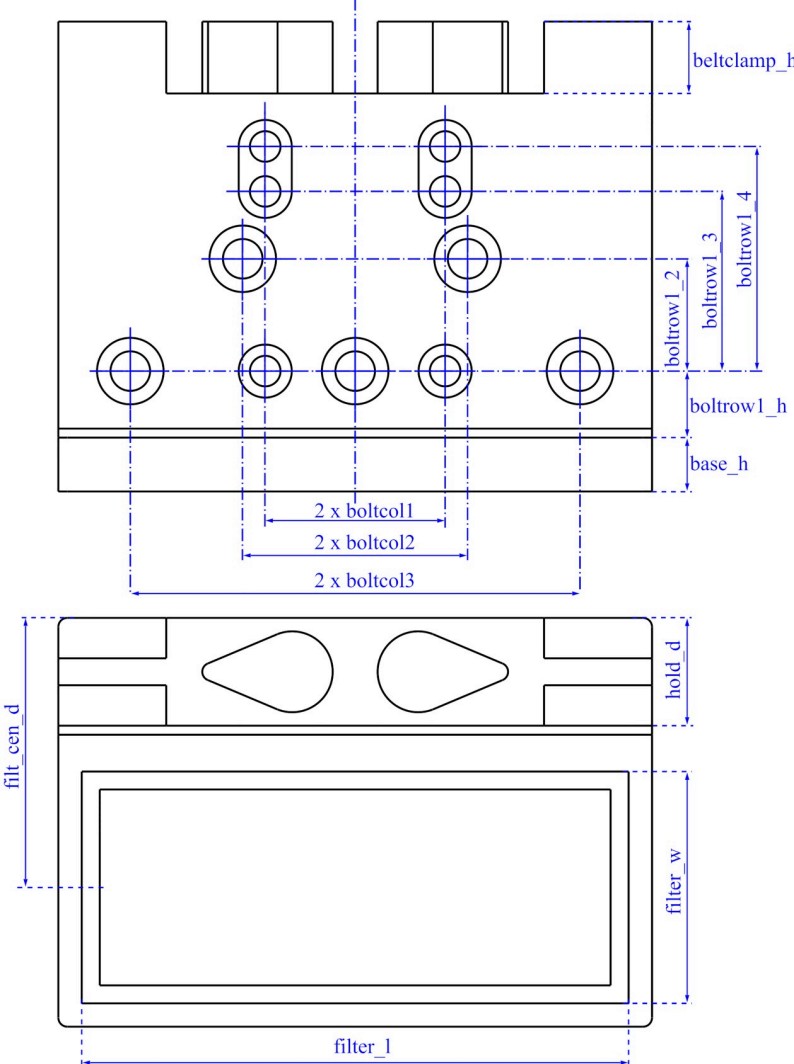

**Fig 10. Main filter holder parameters.**

among the components. Thus, relieving the user from considering these relationships because they are automatically taken into account by the parametric model.

Basically, the tensioner holder depends on the idler tensioner dimensions, and the idler tensioner depends on the idler pulley dimensions. In particular, the parameters that determine the dimensions of the idler tensioner are:

- Idler pulley size.

- Tensioner stroke.

- Thickness of the walls.

- Leadscrew metric.

The parameters that determine the dimensions of the tensioner holder are:

- Idler tensioner size (determined by the aforementioned parameters).

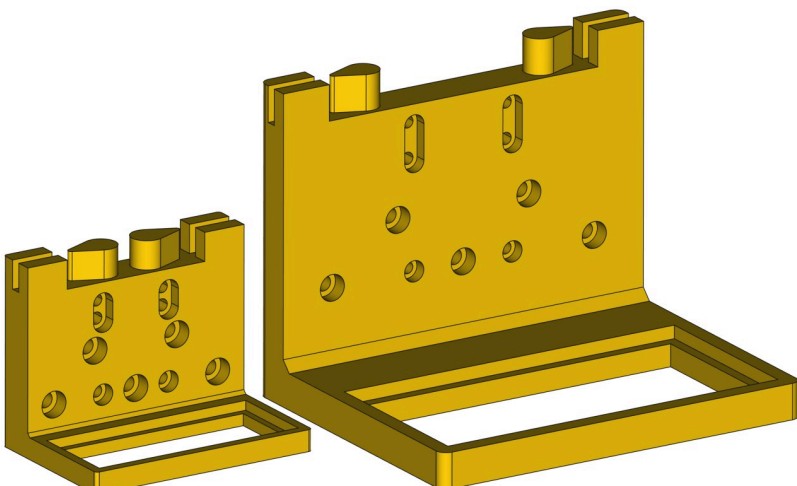

**Fig 11. Two filter holders for different filter sizes and linear guides.**

- Belt height.

- Profile size where it is going to be mounted and its bolts size.

- Thickness of the walls.

Following, these parameters are explained in detail.

**Idler pulley.** Some of the dimensions of the printable pieces depend on the idler pulley size. Idler pulleys can be easily acquired; but also, they can be made using bearings and washers, or they can be 3D printed. Fig 12 shows an example of an idler pulley made of a specific bearing and some washers.

The idler pulley can be made with different component sizes. For example, Fig 13 shows two possible configurations and their influence on the idler tensioner size. The figure shows that the width of the idler tensioner is determined by the pulley size. Also, the space for the pulley will vary depending on the pulley size.

**Tensioner stroke.** The tensioner stroke defines the length of the tensioner; that is to say, how much it can be extended or retracted. Fig 14 shows three different tensioner stroke values and their influence on the tensioner length.

**Wall thickness.** The thickness of the walls has an effect on the overall height of the tensioner (Fig 15). The wall thickness has also a small influence on the length.

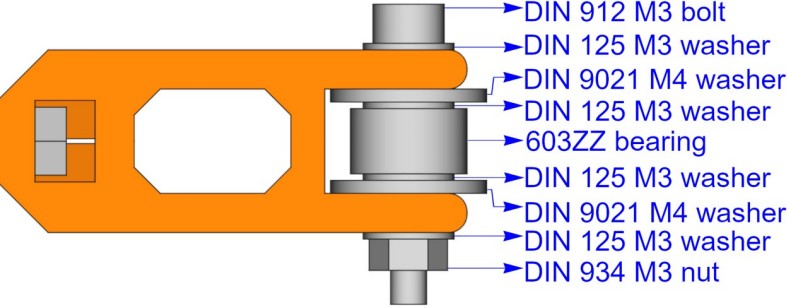

**Fig 12. Example of an idler pulley made out of a bearing and some washers.**

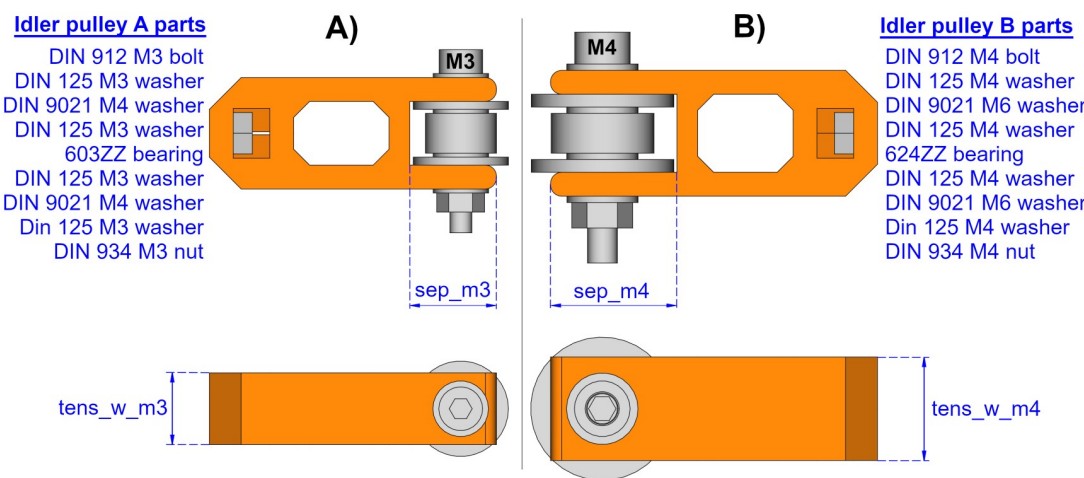

**Fig 13. Comparing two idler tensioners made out of different components.** The size of the idler tensioner is smaller when it contains an idler pulley using a M3 bolt (A), than when using a M4 bolt (B). For example, the space for the pulley or the tensioner width are smaller for case A than case B, as the figure shows that `sep_m3 < sep_m4` and `tens_w_m3 < tens_w_m4`.

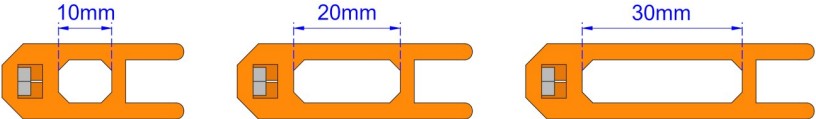

**Fig 14. Idler tensioners with different stroke lengths.**

**Leadscrew diameter.** Depending on the leadscrew diameter, a different size for the nut hole will be needed. This will have a small effect on the idler tensioner length (Fig 16). Obviously, the diameter of the leadscrew hole will also change.

**Idler tensioner size.** All the previous parameters determine the idler tensioner dimensions. Since the idler tensioner is coupled inside the tensioner holder, the idler tensioner size

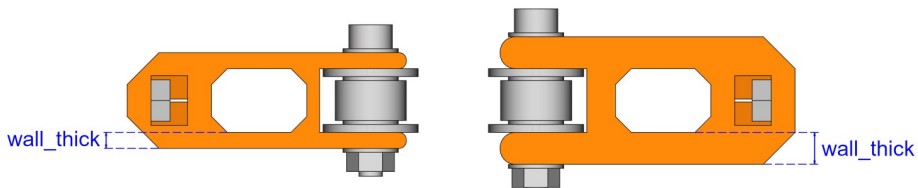

**Fig 15. Idler tensioners with different wall thickness.**

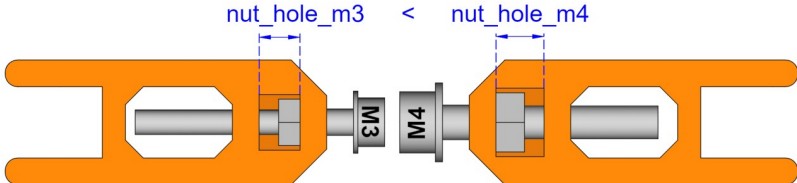

**Fig 16. Idler tensioners with different leadscrew diameters.**

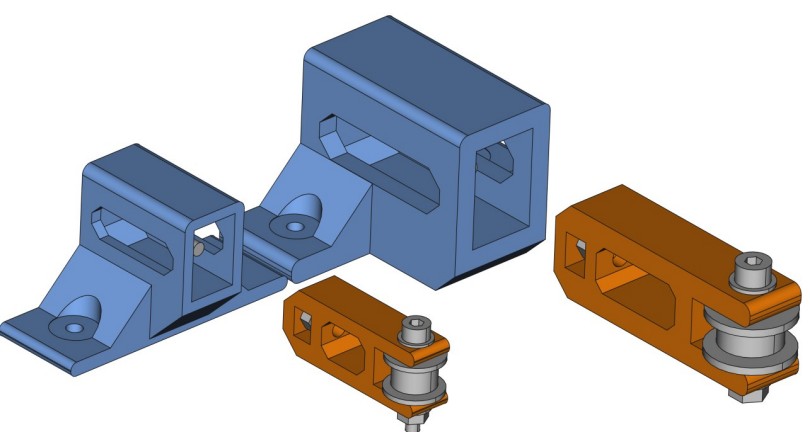

**Fig 17. Tensioner holder dimensions depending on idler tensioner sizes.**

impacts on the size and shape of the tensioner holder. Fig 17 shows two idler tensioners having different parametric values. The corresponding tensioner holders also result in different shapes and sizes. Note how the wall thickness is the same parameter for both the holder and the tensioner.

**Belt height.**   This parameter sets the position of the bottom of the belt along the height of the tensioner (Fig 18A). Changes in this parameter will generate taller or shorter tensioner holders as shown in Fig 18B.

**Base width.**   The base width or profile size indicates the profile size where the belt tensioner will be mounted. Fig 19 shows the resulting belt tensioners for three different profile sizes.

**Other parameters.**   There are other parameters but they do not have a considerable influence over the size. Examples of such parameters are the radius of the fillets or the size of the profile bolts. They are documented in the source code [49] [50].

## Discussion

In this section we compare OpenSCAD and FreeCAD Python. The analysis will include Cad-Query v1.2 as a part of FreeCAD since, as stated above, CadQuery v1.2 can be added as an

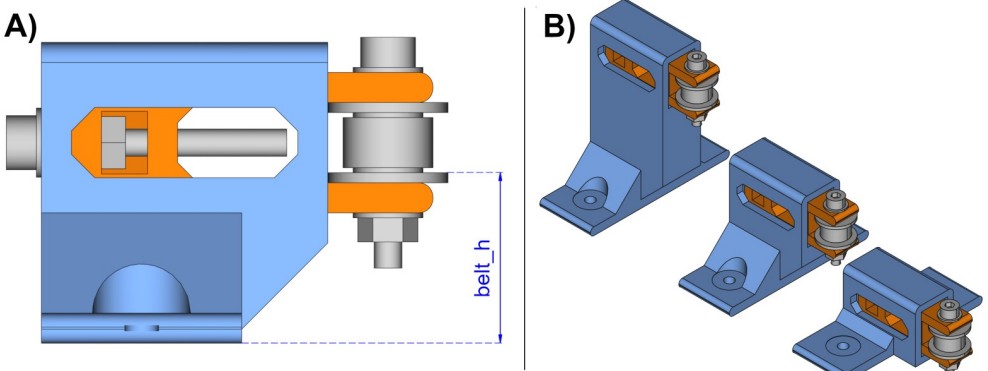

**Fig 18. Belt height parameter.** (A) Definition of the belt height. (B) Belt tensioners with different belt heights.

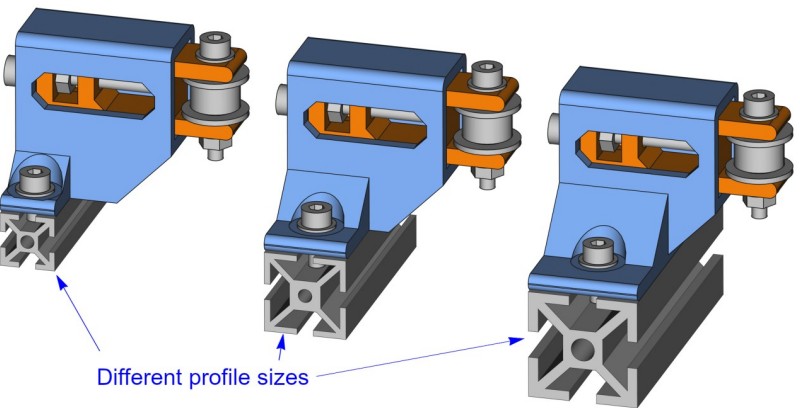

Different profile sizes

**Fig 19. Belt tensioners with different base width.**

external FreeCAD workbench. Throughout this section, we will use the terms CadQuery workbench or just CadQuery to refer to CadQuery v1.2.

We have organized the discussion in four main topics: the geometric modeling kernel, the usability, the programming language characteristics and the tool features.

### Geometric modeling kernel

Both OpenSCAD and FreeCAD use an external library as a geometric modeling kernel.

OpenSCAD uses the CGAL constructive solid geometry library [51]. In constructive solid geometry (CSG) solid models are created by applying successive operations to a set of basic shapes. These shapes are called primitives and the operations can be rigid motions (such as translation and rotation) or boolean operations (union, intersection and difference) [52] [53].

FreeCAD uses the Open Cascade Technology (OCCT) libraries [47]. OCCT is based on boundary representation (B-rep), in which objects are represented by their topological boundaries. A solid model is defined by a set of surface elements that delimit the boundary between the interior and the exterior of the solid. The surface elements are often defined by parametric equations [52] [53].

CadQuery v1.2 is a Python library built on top of FreeCAD API; consequently, it ultimately also uses the OCCT kernel.

One of the most serious disadvantages of OpenSCAD is the use of CGAL because CGAL works with polygonal meshes rather than parametric models. On the other hand, this problem is not present in FreeCAD because OCCT is a parametric modeler.

### Easy of use

The usability has been divided into two subsections: how difficult is to step up the tool to just start coding and how easy is to model with the given language.

**Tool setup.** OpenSCAD is extraordinarily easy to set up. OpenSCAD has an editor window where any code can be tested; thus, it is a matter of start coding and then, clicking a button to save, preview, render or generate the resulting STL file of the model. The code can be saved in a text file with an *.scad extension.

Moreover, OpenSCAD has some model examples that can be loaded from the menu. This is a simple but a very useful characteristic to test the tool capabilities and learn by example.

OpenSCAD allows to have file dependencies in order to keep commonly used functions or constants in a different file. To define the location of these files, the relative path has to be included in the appropriate command.

On the other hand, using Python scripts in FreeCAD is not that simple. Actually, the tutorial about using Python in FreeCAD is placed in a section called "Power users hub" [54], what suggest that FreeCAD is not intended to be used in this way by the novice.

Since modeling in FreeCAD is primarily conceived to be done graphically instead of programmed, there is no code editor window when starting FreeCAD as in OpenSCAD. Instead, we have found two options.

The first option is the Macro Editor, where Python scripts can be executed and saved in the *User macro* directory.

The second option is the Python Console, where any command can be executed; however, it is a console, not a file editor where the user can save their design in a text file, as it is in OpenSCAD. From this console, any Python file can be loaded, but it is not straightforward. Similarly, loading other files dependencies is not direct unless the files are kept inside some specific directories.

The associated FreeCAD Python software repository of the OSH filter stage design [50] includes further information about how to execute the scripts.

Alternatively, the CadQuery workbench can be easily installed in FreeCAD through the Addon Manager, which is available from the graphical user interface. This workbench includes an editor window that allows editing and executing Python code. Furthermore, this workbench has several CadQuery example designs that can be loaded to learn by example.

CadQuery workbench and its editor can be used to design using both CadQuery and FreeCAD APIs. Therefore, it is a good place to start modeling with FreeCAD Python scripts in any of these two options available. Nonetheless, we find that the error messages given through the CadQuery workbench are more obscure than those given through FreeCAD Python console. Thus, we find debugging with CadQuery more difficult than using the FreeCAD console.

**Modeling.** OpenSCAD language is similar to C programming language. OpenSCAD has a few basic 2D and 3D primitives (such as circle, square, polygon, cube, sphere, polyhedron and cylinder) and some operations and transformations. With these basic primitives and operations almost any technical piece can be modeled. Nevertheless, there are some operations that we missed, such as filleting or chamfering.

Since OpenSCAD has a limited set of primitives and functions, it is relatively easy to learn. There are good step-by-step tutorials that provide all the information to become skillful within a short time [21] [55].

As an example, Fig 20A shows the OpenSCAD code to model a box (rectangular cuboid).

FreeCAD uses Python; thus, there is no need to learn a new programming language for those who already know it. FreeCAD offers a Python Application Programming Interface (API) to its OCCT kernel [47]. This API allows creating and accessing OCCT geometric primitives and functions. Essentially, there are two different kind of objects in FreeCAD Python: OCCT shapes and FreeCAD objects. OCCT shapes are the underlying OCCT solid models. On the other hand, FreeCAD objects link the OCCT shapes to their graphical representation. Therefore, in order to display an OCCT shape in FreeCAD graphical interface, there should be a FreeCAD object associated to that OCCT shape.

Fig 20B shows a FreeCAD Python script where these two objects are created. The OCCT shape (`sbox`) is created in line 5. The FreeCAD object is created in line 6; thus, drawing the OCCT shape in the graphical interface. In addition to these objects, the code include lines 2 and 3 to import libraries, and line 4 to create a FreeCAD document to be able to save it. Once

```
1  // openscad
2  cube([10,20,30]);
                                    A)
```

```
1  # FreeCAD Python                          B)
2  import FreeCAD
3  import Part
4  doc = FreeCAD.newDocument()
5  sbox = Part.makeBox(10,20,30)
6  Part.show(sbox)
```

```
1  # CadQuery FreeCAD Python                  D)
2  import cadquery as cq
3  import Helpers
4  cqbox = cq.Workplane("XY").box(10,20,30)
5  Helpers.show(cqbox)
```

```
1  # FreeCAD Python                           C)
2  import FreeCAD
3  doc = FreeCAD.newDocument()
4  fbox = doc.addObject("Part::Box","MyBox")
5  fbox.Length = 10
6  fbox.Width  = 20
7  fbox.Height = 30
```

**Fig 20. Sample codes to model a rectangular cuboid.** Code (A) is modeled in OpenSCAD. Codes (B) and (C) are modeled using FreeCAD Python scripts. Code (D) is modeled using a Python script for FreeCAD CadQuery workbench.

the FreeCAD document is saved, not only the source code can be shared but also the FreeCAD model, allowing it to be modified through the FreeCAD graphical interface.

Alternatively, FreeCAD API offers functions that automatically create FreeCAD objects with its underlying OCCT shape. These functions hide the OCCT shape from the user. Fig 20C shows an example where a FreeCAD object (fbox) is created in line 4. This FreeCAD object already links to its OCCT shape, which can be accessed through an attribute. Lines 6 to 8 are used to assign the box dimensional values.

Although there are some tutorials about FreeCAD Python scripting [54], they are not comprehensive and the documentation is neither complete nor well organized. However, since FreeCAD GUI commands are just Python scripts, and these scripts can be redirected to FreeCAD Python console, an alternative way to learn is to create models with FreeCAD GUI commands and observe their corresponding Python scripts in the console.

CadQuery v1.2 is a Python library on top of the FreeCAD API. CadQuery eases parametric feature-based modeling by providing methods to facilitate the location and creation of features. Nevertheless, although the design approach is intuitive for simple or symmetrical models, we have found that it may be confusing for intricate pieces. CadQuery has well organized documentation and tutorials, what improves the learning process [56].

CadQuery creates its own type of objects for solid modeling. Since CadQuery is a library on top of the FreeCAD API, CadQuery objects also link to the underlying OCCT shapes. In addition, if we want to represent the shapes in the FreeCAD GUI, a FreeCAD object has also to be created.

Fig 20D shows an example of a CadQuery script. A CadQuery object is created in line 4. This object includes the underlying OCCT shape. Line 5 creates the FreeCAD object that draws the shape in FreeCAD GUI, although the user may not realize that a FreeCAD object has been created. Line 4 of Fig 20D exposes a differentiated CadQuery characteristic: CadQuery builds the geometry from defined planes (Workplane).

As a summary, Fig 20 shows different codes to model a rectangular cuboid. The three FreeCAD Python examples (B, C, D) contrast with OpenSCAD simplicity (A) and illustrate the initial difficult that the newcomer may face using FreeCAD scripts. He will not only need to learn how to describe a model using a programming language, but also he would have to understand the intricacies of the different objects, methods, libraries and the underlying OCCT kernel. It may not be a problem if the designer has a programming background, but if the designer has little experience in programming, the initial barrier could be high.

```
1  x = 5;
2  sphere(x);          assignment
3  x = 10;             with no effect
4  sphere(x);    A)
```

```
x = x + 1; ☒    invalid
                assignment
           B)
```

**Fig 21. OpenSCAD sample codes to show the effect of its functional programming paradigm.** (A) Assignment in line 1 has no effect since variables keep a constant value during their entire lifetime. (B) The code shows an invalid assignment since variables cannot change their values.

## Programming language characteristics

In this subsection, some of the most significant programming language characteristics of both tools are compared. The analysis includes the programming paradigm, the scope of variables, the data types and the libraries.

**Programming paradigm.** Programming languages can be classified based on their features. OpenSCAD is a declarative, purely functional language [55]; whereas Python supports multiple programming paradigms, including procedural, object-oriented, and functional programming [34].

As a consequence of OpenSCAD functional programming paradigm, variables are set at compile time, not at run time. Therefore, variables keep a constant value during their entire lifetime. If a variable is assigned a value multiple times, only the last value is used in all places of the code. This characteristic can be confusing for programmers used to procedural languages such as C or Python. For example, in Fig 21A variable x will have a constant value of 10; consequently, this script will create two spheres of the same radius 10. The assignment in line 1 has no effect and may mislead the designer in thinking that the spheres will have different size. The sphere created in line 2 will have a radius of 10 even that the final assignment of variable x is made afterwards (line 3).

For this same reason, the assignment x = x + 1 is not valid in OpenSCAD (Fig 21B).

On the other hand, Python is a multi-paradigm language and variables can be modified at running time as in procedural languages. Furthermore, it can make use of other paradigms such as object oriented programming, which is extensively adopted in FreeCAD Python.

Functional languages have benefits such as being more predictable and less prone to bugs; however, the programmer used to procedural paradigms may find functional programming too rigid to make fully parametric designs.

**Scope of variables.** Variables are created within a scope in OpenSCAD; thus, their values are not available outside that scope. Fig 22A shows a situation where the variable scope may produce a different behavior than expected. Since y is assigned both outside and inside the if statement, y will have two different values depending on the scope. As a consequence, the

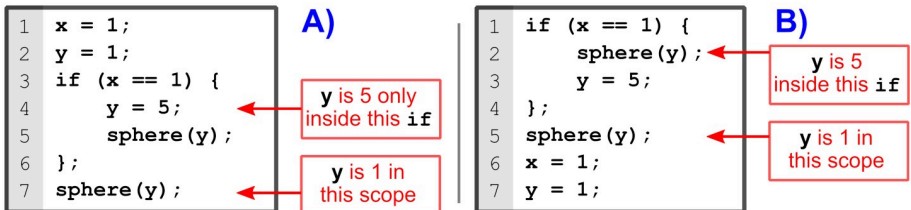

**Fig 22. Sample codes to show the variable scope in OpenSCAD.** (A) Assignment in line 4 has no effect outside the if statement; therefore the sphere in line 7 will have a radius of 1. (B) This code produces the same result as in (A) even that the variable assignments are placed after the sphere function calls that use those variables.

```
1    r = 10;                    A)
2    if (r == 5) {
3        h = 15;
4    } else {
5        h = 20;
6    }
7    cylinder(r=r,h=h);
```
h is 15 only inside this **if**
h is 20 only inside this **else**
h is undefined outside the **if**

```
1    r = 10;                    B)
2    if (r == 5) {
3        h = 15;
4        cylinder(r=r,h=h);
5    } else {
6        h = 20;
7        cylinder(r=r,h=h);
8    }
```

```
1    r = 10;                          C)
2    h = (r == 5) ? 15 : 20;
3    cylinder(r=r,h=h);
```

**Fig 23. Parametric design and variable scope in OpenSCAD.** Code (A) will not work because h has not been defined outside the if statement. Codes (B) and (C) will work.

sphere in line 7 will not be affected by the assignment of y inside the if statement (line 4); thus, producing spheres of different size.

Note how scripts in Fig 22A and 22B are equivalent. The differences lie in the assignments placement, but since OpenSCAD is a functional language, both descriptions generate the same model.

OpenSCAD scoping rules can make parametric models more complicated to design when establishing variable dependencies. For example, suppose we want to create a cylinder whose height depends on its radius value. If the radius is 5, the height will be 15; otherwise, the height will be 20. Fig 23 shows shows three attempts to do this.

The first attempt (Fig 23A) will not work because h has not been assigned outside the if statement; therefore, it will be undefined when calling the cylinder function.

The second script (Fig 23B) has solved the problem by creating the cylinder inside the if alternatives. Nevertheless, this solution lacks efficiency because the same function call has to be repeated. For this small example we had to repeat the cylinder function call in lines 4 and 7.

The third example (Fig 23C) uses the conditional ? to avoid assigning h in an inner scope. Although we can resort to the conditional ? to create variables in an outer scope, it can be cumbersome to use it when there are many alternatives or when there are more than one variable to assign. We have experienced this problem when defining the dependencies of the belt tensioner, as it can be seen in OpenSCAD file kidler.scad [49]. Nonetheless, it can be used efficiently in combination with vectors, as some libraries of technical components have done [57] [58] [59].

Python does not have this behavior. Variables can be updated anywhere in the code, and they keep their value within their scope. Unlike OpenSCAD, the programmer can define the scope of variables using local, global and nonlocal variables.

**Data types.** OpenSCAD has a limited set of data types: number (64 bit floating point), boolean, string, range, vectors and undefined. There are not user defined types in OpenSCAD.

In contrast, Python has several data types. Python provides the standard built-in data types, but also other specialized data types defined in the Python standard library and other available modules. In addition, programmers can create their own data types.

The standard data types can be summarized in numbers (integer, floating point and complex), strings, lists, tuples, sets and dictionaries. Some of them can be very useful for managing information, like lists and dictionaries. We have extensively used them to define the dimensions of the components of our OSH test-bench.

For example, Fig 24A shows how a dictionary can be used to get the thickness of a DIN 125 washer. In Fig 24B a two-dimensional dictionary is used.

It is possible to have relatively analogous structures in OpenSCAD to get same result. Like using the conditional ? with vectors (similar to Fig 23C). However, Python data types allow

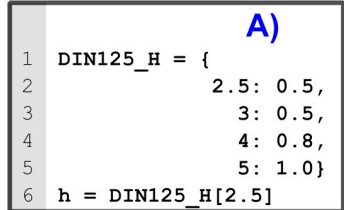

**Fig 24. Sample codes of dictionaries for defining component dimensions in Python.** (A) Dictionary DIN125_H defines the height (thickness) of some DIN 125 washers. Keys can be a float number, as in line 2. Line 6 shows how to obtain the height (0.5) of a DIN125 M2.5 washer from the dictionary. (B) Dictionaries can be multidimensional and can have strings as keys.

much more flexibility; moreover, Python offers a set of optimized methods and functions to manipulate these data structures efficiently.

**Libraries.**  OpenSCAD has a limited set of functions; however, anyone can create libraries to facilitate the design process. OpenSCAD has some libraries available ranging from mathematical functions to the creation of useful shapes and mechanical parts [57] [58] [59] [60].

FreeCAD has libraries of components [57] [61] and also has modules to extend FreeCAD functionality, such as CadQuery [42]. These modules provide a wide set of tools varying from advanced modeling to the creation of all kind of mechanical objects. The modules are available through the graphical interface as workbenches [62] [63].

In addition to these specific FreeCAD extensions, the FreeCAD Python programmer can resort to the rich and versatile Python standard library and other specialized modules. Python libraries are developed by a vast community, much larger than the FreeCAD or OpenSCAD specific communities. This is of paramount importance because it provides innumerable standardized solutions to FreeCAD Python.

As an example, file manipulation in FreeCAD Python is made through Python libraries, not because FreeCAD developers implemented that functionality. Consequently, if a FreeCAD Python designer wants to include the ability to read data from a file, she would just need to use the appropriate Python library to read and parse that file. In contrast, adding the ability to read and manipulate files in OpenSCAD would require OpenSCAD developers to implement that functionality.

This characteristic allows Python programmers to count on libraries for innumerable tasks. Examples of this kind of tasks are reading text files to get parameter values; writing text files to generate reports or bill of materials; performing many kind of computations, such as mathematical, matrix and finite elements; error handling; working with data interchange formats, like JSON, XML, YAML; among many other tasks. As a result, Python libraries add a broad range of capabilities to FreeCAD Python.

## Tool features

Both OpenSCAD and FreeCAD are multiplatform (Windows, MacOS and Linux) free and open source software. OpenSCAD is under GPL2 and FreeCAD under LGPL2+.

This subsection analyzes two important features: the graphical interface and the import/export capabilities. In addition, we explore some characteristics like the performance and the suitability for modeling complete systems. To end, we look into the status of the software project development.

**Graphical user interface (GUI).**  OpenSCAD GUI has commands for visualization such as zoom, rotation, change of perspective, among a few other commands. There is the option to include axis and perform animations. OpenSCAD also allows changing the color and

transparency of the objects, although it cannot be done through the GUI, but must be defined in the code. OpenSCAD also features a parameter customizer that allows changing the model parameters from the GUI.

On the other hand, since FreeCAD has been mainly devised to be used through its GUI, it has all kind of commands to visualize, measure and transform the models. Commands are organized in workbenches to perform related tasks. As an example, it has the Technical Draw Workbench, which produces basic technical drawings of the models similar to Figs 8 and 10.

Another remarkable feature of FreeCAD is its ability to integrate the scripts as commands in the graphical interface, allowing designers to generate and customize the models through the GUI. Therefore, regular users would not need to work with the source code. As a conclusion, FreeCAD GUI is clearly more powerful and flexible than OpenSCAD.

**Import/Export capabilities.**   As we have said, OpenSCAD has limited export/import capabilities. OpenSCAD can import some 2D file formats but only imports 3D tessellated file formats such as STL. The problem with these file formats is that they no longer contain parametric information.

This is an important limitation when working in projects with different CAD tool users. The OpenSCAD designer cannot incorporate exact models from other CAD tools in her/his design.

Likewise, OpenSCAD can only export to tessellated 3D file formats. These formats are suitable for 3D printing, but not to get an exact representation of the model. From our point of view, this is one of the main OpenSCAD drawbacks, since it inhibits sharing exact dimensional models. It seems that OpenSCAD is more oriented to the creation of models for 3D printing, but not to create exact dimensional models of the pieces that can be used in other CAD tools.

Actually, as it has been said, the question is that OpenSCAD uses a kernel that works with polygonal meshes rather than parametric models. Therefore, despite that the design have been modeled parametrically in OpenSCAD, the parametric information is lost once the internal model is generated.

Fig 25 visually explains this situation with an example. On the left we have the source codes for OpenSCAD (Fig 25A) and FreeCAD (Fig 25B). OpenSCAD can only generate 3D tessellated models, which can be exported to mesh file format such as a STL. As it can be observed, the tessellated model is made of a polygonal mesh, what implies that the parametric dimensions, such as the radius of the primitives (sphere, cone and cylinder), have been lost.

On the other hand, FreeCAD generates a parametric model that can be exported to a standard parametric file format such as STEP. This file format preserves its original dimensional information and can be manipulated by most of CAD tools; that is to say, it can be used as a source for future modifications in other CAD tools. This is an important feature, since other potential users may not be interested in coding, or they use a CAD tool that cannot import the source code. In addition, FreeCAD can also generate the tessellated model used for production.

In addition to FreeCAD ability to import/export to standard parametric models, FreeCAD has a workbench to offer interoperability with OpenSCAD. This workbench contains functions to import and repair OpenSCAD models. Nevertheless, depending on the OpenSCAD primitives used, some of the parametric information may be lost.

**Speed.**   We have attempted to compare the time taken to generate the models in both tools. However, since OpenSCAD creates a tessellated model, the speed depends on the mesh resolution. The resolution can be defined by OpenSCAD special variables $fa, $fs and $fn. Variable $fa defines the minimum angle for a fragment, $fs defines the minimum size of a fragment.

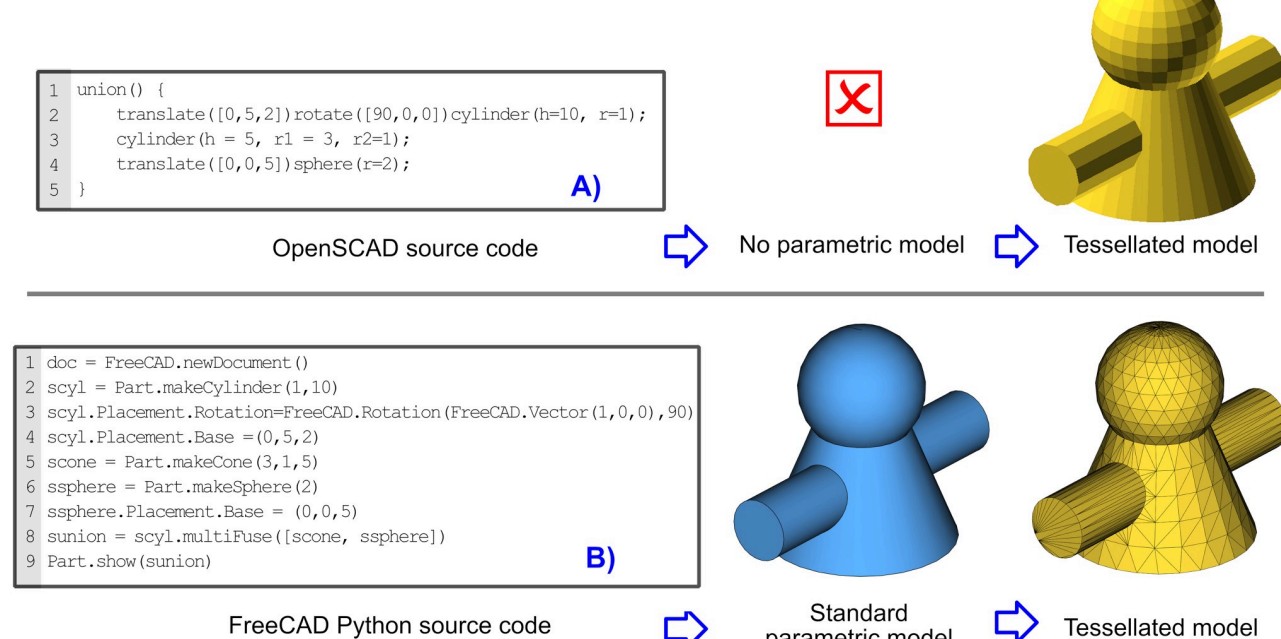

**Fig 25. Source codes and their generated models.** (A) OpenSCAD can export to polygonal mesh models, but not export to parametric models. (B) FreeCAD can export to both polygonal mesh models and standard parametric models.

On the other hand, there is no need to specify the mesh resolution in FreeCAD unless a tessellated model is needed. Table 2 shows the time in seconds that took FreeCAD to generate the parametric models. FreeCAD CadQuery models take a similar time than plain FreeCAD Python because they use the same FreeCAD API. Actually, some of CadQuery models used in this study took slightly less time because, in order to make the tutorials, the FreeCAD Python models include the graphical representation of every step of the construction process. Nevertheless, since the execution times are very similar, for clarity purposes, only the execution time of the plain FreeCAD Python models will be included.

In order to compare the speed, all the scripts have been executed in a computer with an Intel Core i7-4700Q CPU at 2.4GHz and 8 GB RAM running 64 bit Windows 10 Home. In FreeCAD Python the completion time has been measured with the function `datetime.now()` (code is available in [50]). In OpenSCAD the completion time has been obtained by the provided rendering report. This report does not provide fractions of seconds.

The tessellated models generated by both tools are different, as it can be appreciated in Fig 25. We have not found any option in OpenSCAD to generate a different kind of mesh.

**Table 2. Time to generate parametric models.**

| Parametric time | OpenSCAD | FreeCAD |
|---|---|---|
| Motor bracket | * | 0.43 s |
| Filter holder | * | 0.76 s |
| Idler tensioner | * | 0.45 s |
| Tensioner holder | * | 0.61 s |

* OpenSCAD does not generate parametric models.

**Table 3. Selected parameter values to get different mesh resolutions.**

|  | OpenSCAD | | FreeCAD | |
|---|---|---|---|---|
|  | `$fa` | `$fs` | `ad` | `sd` |
| Coarse | 6° | 0.4 mm | 30° | 0.1 mm |
| Normal | 6° | 0.2 mm | 15° | 0.05 mm |
| Fine | 3° | 0.1 mm | 3° | 0.01 mm |

FreeCAD has three meshing options: standard, Mefisto and Netgen. Each option has their own parameters to adjust the resolution and sometimes the shape of the mesh. FreeCAD standard mesh seems to be the most similar to OpenSCAD mesh. FreeCAD standard mesh has two parameters to define the mesh resolution: the angular deviation (`ad`) measured in angular degrees, and the deviation of the surface (`sd`) measured in millimeters. The former limits the angle between subsequent segments in a polyline. The latter limits the distance between a curve and its tessellation [64]. Note that these parameters do not correspond to OpenSCAD special variables; although in both cases, the smaller the values are, the higher the resolution will be.

Starting from values that in our experience have provide a good 3D printing quality, we have increased the resolution of both meshes in order to have more data to compare. We have defined three levels of coarseness: coarse, normal and fine. Table 3 shows the chosen values for these parameters.

Table 4 shows the number of elements (vertices, edges and facets) of each of the generated meshes of the motor bracket. From the table we can see that the meshes generated by Open-SCAD and FreeCAD are different, since for a similar number of vertices, FreeCAD meshes have a much larger number of edges and facets than OpenSCAD meshes.

FreeCAD is clearly faster generating meshes, especially when the resolution is increased. OpenSCAD times rise at a much higher rate for finer resolutions. At the finest resolution Free-CAD is more than ten times faster.

FreeCAD generates the meshes from the parametric model. Therefore, once the parametric model has been generated, the parametric time (Table 2) can be subtracted from the total time (Table 4). Comparing the time values of these tables, it can be observed that FreeCAD only uses a fraction of the total time to generate the mesh of the motor bracket.

Table 5 shows the same information for the filter holder. The filter holder is the most complex piece. As a result, it has the largest amount of elements and it takes more time to generate the meshes. Again, it can be observed that for larger meshes FreeCAD is much faster than OpenSCAD. For the fine resolution is more than 50 times faster.

The idler tensioner and the tensioner holder are not intricate pieces. Their complexity lies on their parameters dependencies, but once the pieces are created, the meshes are relatively

**Table 4. Motor bracket: Number of vertices, edges and facets of the meshes and time to generate them.**

| Motor bracket | OpenSCAD | | | FreeCAD | | |
|---|---|---|---|---|---|---|
|  | Coarse | Normal | Fine | Coarse | Normal | Fine |
| Vertices | 479 | 807 | 1,568 | 402 | 722 | 3,394 |
| Edges | 719 | 1,211 | 2,352 | 1,242 | 2.202 | 10,218 |
| Facets | 242 | 406 | 786 | 828 | 1,468 | 6,812 |
| Total time [a] | 2 s | 4 s | 8 s | 0.45 s | 0.47 s | 0.57 s |

[a]FreeCAD time includes the generation of the parametric model (Table 2)

**Table 5. Filter holder: Number of vertices, edges and facets of the meshes and time to generate them.**

| Filter holder | OpenSCAD | | | FreeCAD | | |
|---|---|---|---|---|---|---|
| | **Coarse** | **Normal** | **Fine** | **Coarse** | **Normal** | **Fine** |
| Vertices | 1,972 | 2,746 | 5,422 | 1,178 | 2,282 | 11,114 |
| Edges | 2,958 | 4,119 | 8,133 | 3,600 | 6,912 | 33,408 |
| Facets | 1,000 | 1,387 | 2,725 | 2,400 | 4.608 | 22,272 |
| **Total time** [a] | 30 s | 45 s | 102 s | 0,87 s | 0,90 s | 1,47 s |

[a]FreeCAD time includes the generation of the parametric model (Table 2)

simple. (Table 6) shows their meshes size and the time to create them. The resulting data is in line with of these tables support the previous analysis.

Fig 26 summarizes the total time to generate the meshes. As can be seen in the graph, the worst case in FreeCAD (filter holder fine mesh) is slightly faster than the best case in Open-SCAD (motor bracket coarse mesh). Nevertheless, given that the meshes are not equivalent and the limited number of pieces used in the study, the results should be treated with caution.

**System modeling.** A complete mechanical system can be modeled with OpenSCAD and there are some very good examples in OpenSCAD gallery [65]. Nevertheless, we believe that OpenSCAD is more oriented to create solid models of individual pieces with the purpose of 3D printing. We find some arguments to support this idea.

First, a system model is useful to get a visual idea, document and share the complete device. However, OpenSCAD lacks a powerful graphical interface and no model transformations can be done through it.

Secondly, OpenSCAD import/export capabilities are limited to tessellated file formats; thus, OpenSCAD models cannot be integrated in a design of a different CAD tool without losing information. Conversely, a parametric model created in other CAD tool cannot be integrated in a OpenSCAD design without information loss.

An example of the relevance of the import/export capabilities for generating a system model is the collaboration between Electronic CAD (ECAD) and Mechanical CAD (MCAD) tools to generate models that integrate electrical and mechanical components [66]. FreeCAD

**Table 6. Idler tensioner: Number of vertices, edges and facets of the meshes and time to generate them.**

| Idler tensioner | OpenSCAD | | | FreeCAD | | |
|---|---|---|---|---|---|---|
| | Coarse | Normal | Fine | Coarse | Normal | Fine |
| Vertices | 474 | 830 | 1,610 | 316 | 580 | 2,694 |
| Edges | 713 | 1,247 | 2,417 | 972 | 1,764 | 8,106 |
| Facets | 241 | 419 | 809 | 646 | 1,174 | 5,400 |
| **Total time** [a] | 4 s | 7 s | 13 s | 0.54 s | 0.54 s | 0.61 s |
| Tensioner holder | OpenSCAD | | | FreeCAD | | |
| | Coarse | Normal | Fine | Coarse | Normal | Fine |
| Vertices | 535 | 778 | 1,498 | 322 | 591 | 2,741 |
| Edges | 777 | 1,142 | 2,192 | 990 | 1,797 | 8,247 |
| Facets | 246 | 368 | 698 | 660 | 1,198 | 5,498 |
| **Total time** [a] | 6 s | 8 s | 16 s | 0.63 s | 0.64 s | 0.71 s |

[a]FreeCAD time includes the generation of the parametric model (Table 2)

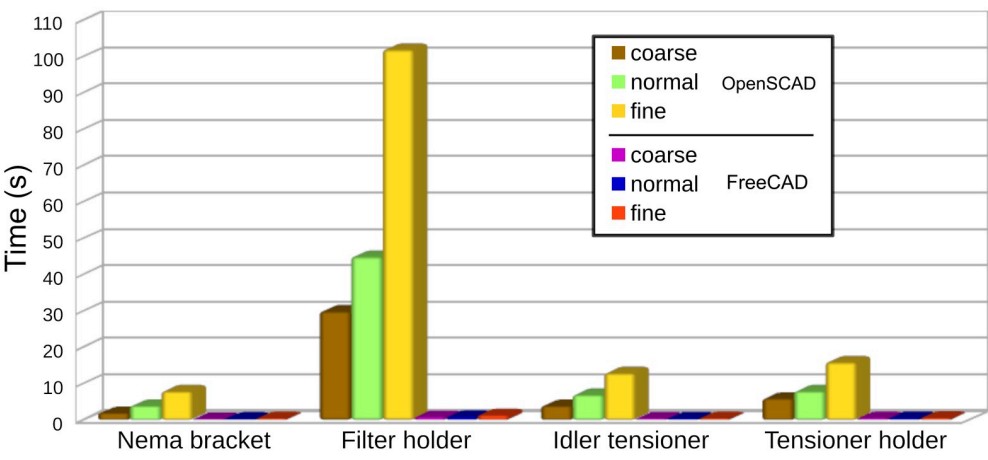

**Fig 26. Mesh generation times in OpenSCAD an FreeCAD.**

parametric models of electronic components can be exported to electronic design tools such as KiCad [67]. As a result, KiCAD can generate the whole CAD model of the electronic board with its components. Finally, the resulting CAD model of the board could be integrated into FreeCAD in order to create the CAD model of the whole electromechanical system. There is even a FreeCAD workbench aiming to foster the collaboration between KiCad and FreeCAD [68]. Although there is a way to do it in OpenSCAD, it is not direct and the parametric information is lost.

Thirdly, OpenSCAD rendering speed slows in complex models, what supports the idea that it is not a tool to generate a whole system.

In contrast, FreeCAD is conceived to model both individual pieces or a complete system. It has a powerful GUI and it is mainly developed to be used in this way, but at the same time, it has powerful programming capabilities. The generated models can be exported and imported into standard formats, so CAD files can be incorporated and shared to the public without losing information. As an example, the complete parametric OSH system presented in this article (Fig 1) has been modeled using FreeCAD Python.

**Tool development.** To end this subsection we analyze both projects in terms of their development. The objective is to compare the data publicly available in an attempt to obtain some metrics in order to characterize the development. We have chosen some of the metrics suggested by Crowston and Howison [69], such as the number of developers (contributors), the level of activity (commits), cycle time (releases) and popularity (downloads). We also include the number of lines of code to offer a intuitive metric of the project size. The information have been gathered from [70] [71] [72] and the projects website, although we could not find download statistics for OpenSCAD.

We observe from Table 7 that although both projects have a high level of activity, FreeCAD is notably more active and have frequents releases. That could be predicted since a graphical CAD tool would attract more users than a scripting CAD tool. However, from the analysis of the contributions on parametric OSH for scientific equipment, we would think that there probably are many more users of OpenSCAD than FreeCAD Python scripts.

## Summary

To recapitulate, OpenSCAD is easy to setup and its modeling language hides the low level details related to the internal data structures and their interaction with the geometry kernel.

**Table 7. General project metrics.**

| Metric | OpenSCAD | FreeCAD |
|---|---|---|
| Lines of code[a] | 129,641 | 3,181,812 |
| Commits during last year[b] | 1,327 | 3,469 |
| Contributors during last year[b] | 32 | 112 |
| Latest release | May 2019 | (0.18.3) July 2019 |
| Latest major release | May 2019 | (0.18.0) Mar. 2019 |
| Previous major release | Mar. 2015 | (0.17.0) Apr. 2018 |
| Downloads of latest major release[c] | N/A | (0.18) 1,298,693 |
| Downloads of previous major release[c] | N/A | (0.17) 1,698,323 |

[a]Not including comment lines or blank lines.

[b]From Sep. 2018 to Sep. 2019.

[c]Including all minor releases. Downloads until Sep. 25th, 2019.

Consequently, OpenSCAD is accessible to the non-expert programmer and allows designers to focus on the CAD modeling.

However, OpenSCAD functional programming paradigm and its scoping rules may seem cumbersome for those who are not used to them. Nonetheless, OpenSCAD main drawback is that its kernel works with tessellated models, in which the parametric geometry is lost. As a result, the interoperability with other CAD tools is dramatically reduced since it is not able to export to standard parametric formats.

On the other hand, FreeCAD Python users need a programming language background to deal with the setup and the intricate programming structures that are exposed. However, installing FreeCAD CadQuery workbench allows FreeCAD users to avoid the setup difficulties and provides them with a friendlier coding interface. Once this barrier is overcome, the Free-CAD Python designer can enjoy a design tool that provides a plethora of advantages, such as exporting to standard parametric formats, using CadQuery library and any other Python libraries, and incorporating the scripted models into its graphical user interface, among many others.

Finally, Table 8 outlines the comparison of both tools.

## Conclusions

Designing open source scientific hardware using a programming language offers two clear benefits: it allows parametric design and provides a source code for the hardware. Parametric is design is a highly desirable characteristic for open source labware because it enables customization to suit different experiment purposes. On the other hand, providing a source code for the hardware helps to make the hardware truly open by mending the lack of enough documentation that OSH have.

Designing open source hardware using Python for FreeCAD has distinct advantages over OpenSCAD. We consider that the longer learning curve of Python for FreeCAD is largely compensated by three major benefits. First, the ability to export to standard parametric CAD formats. Secondly, the usage of a widespread programming language with an extensive standard library. Lastly, the ability to use and integrate the generated models and the scripts in the FreeCAD graphical interface; thus, allowing non-programmers designers to use and configure the models.

**Table 8. Tool characteristics summary.** The table is divided in the four topics: (1) geometric modeling kernel; (2) easy of use, (3) programming languages characteristics and (4) tool features.

| Topic | OpenSCAD | FreeCAD Python |
|---|---|---|
| KERN | Computational Geometry Algorithms Library (CGAL) | Open Cascade Technology (OCCT) |
| | Based on Constructive Solid geometry (CSG) | Based on boundary representation (B-rep) |
| | Polygonal mesh model | Parametric model (B-rep) |
| EASY | Usable, easy to setup | Complex setup—Improved with CadQuery workbench |
| | Easy to learn: reduced set of primitives and functions | Complex to learn: many kind of functions and objects |
| | Easy to learn: higher abstraction level | Complex to learn: dealing with kernel structure |
| | Step-by-step thorough tutorials | Few scripting tutorials—Good tutorials for CadQuery |
| PROG | Parametric design, code based, version control | Parametric design, code based, version control |
| | Specific programming language | Widespread programming language (Python) |
| | Declarative, purely functional paradigm | Multi-paradigm, including procedural and object-oriented |
| | Cumbersome scoping rules | Scope of variables can be defined |
| | Data types: number, boolean, string, range, vector | Whole range of Python data types and user defined |
| | Specific OpenSCAD libraries | Specific FreeCAD libraries |
| | No libraries other than from OpenSCAD | Python libraries provide solutions to countless problems |
| | Not good at handling input/output text files | All kind of functions to read/write text files |
| | No fillet or chamfer transformations | Fillet and chamfer transformations |
| | Minkowski and Hull transformations | No Minkowski or Hull transformations |
| TOOL | Multiplatform FOSS under GPL2 | Multiplatform FOSS under LGPL2+ |
| | No modifications through GUI | All kind of transformations through the GUI |
| | No script integration in GUI | Scripts can be integrated in the GUI |
| | No import/export to standard parametric formats | Import/export to standard parametric formats |
| | Slow model generation | Fast model generation |
| | Difficult for system modeling | Suitable for system modeling |
| | Vibrant community creating OSH scientific equipment | Almost no one creating open source labware |
| | Active software development project | Very active software development project |

KERN, geometric modeling kernel; EASY, Easy of use; PROG, programming language characteristics; TOOL, tool features.

Positive features are shaded in light blue. Negative features are shaded in light red. Non shaded cell describe neutral features or that could be considered positive or negative. Negative FreeCAD features that improve when using the CadQuery workbench are shaded yellow.

In the light of these clear benefits, we hope that our analysis and companion step-by-step tutorials will encourage the scientific community to adopt Python for FreeCAD for modeling parametric open source scientific equipment.

Future work will involve (1) the creation of modules to isolate the CAD designer from the lower level kernel structures; (2) the integration of the scripted models in the FreeCAD graphical interface to allow non-programmers to easily parametrize the designs; and (3) exploration of methodologies to aid parametric system modeling.

## Author Contributions

**Conceptualization:** Felipe Machado.

**Data curation:** Felipe Machado.

**Formal analysis:** Felipe Machado.

**Funding acquisition:** Norberto Malpica, Susana Borromeo.

**Investigation:** Felipe Machado.

**Methodology:** Felipe Machado.

**Project administration:** Norberto Malpica, Susana Borromeo.

**Resources:** Felipe Machado, Norberto Malpica, Susana Borromeo.

**Software:** Felipe Machado.

**Supervision:** Norberto Malpica, Susana Borromeo.

**Validation:** Felipe Machado.

**Visualization:** Felipe Machado.

**Writing – original draft:** Felipe Machado.

**Writing – review & editing:** Felipe Machado, Norberto Malpica, Susana Borromeo.

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
