## [Decision Letter · Decision Letter 0]

12 Sep 2019

PONE-D-19-22759

Parametric CAD modeling for open source scientific hardware: comparing OpenSCAD and FreeCAD Python scripts

PLOS ONE

Dear Mr Machado,

Thank you for submitting your manuscript to PLOS ONE. After careful consideration, we feel that it has merit but does not fully meet PLOS ONE’s publication criteria as it currently stands. Therefore, we invite you to submit a revised version of the manuscript that addresses the points raised during the review process.

To facilitate a timely publication, your revised manuscript should be uploaded as soon as possible.

We would appreciate receiving your revised manuscript by Oct 27 2019 11:59PM. To enhance the reproducibility of your results, we recommend that if applicable you deposit your laboratory protocols in protocols.io, where a protocol can be assigned its own identifier (DOI) such that it can be cited independently in the future. For instructions see: http://journals.plos.org/plosone/s/submission-guidelines#loc-laboratory-protocols

We look forward to receiving your revised manuscript.

Kind regards,

Talib Al-Ameri, Ph.D

Academic Editor

PLOS ONE

Journal Requirements:

3.  We note that Figures in your submission contain copyrighted images. All PLOS content is published under the Creative Commons Attribution License (CC BY 4.0), which means that the manuscript, images, and Supporting Information files will be freely available online, and any third party is permitted to access, download, copy, distribute, and use these materials in any way, even commercially, with proper attribution. For more information, see our copyright guidelines: http://journals.plos.org/plosone/s/licenses-and-copyright.

1.         You may seek permission from the original copyright holder of Figure(s) to publish the content specifically under the CC BY 4.0 license.

Reviewers' comments:

Reviewer's Responses to Questions

**Comments to the Author**

1. Is the manuscript technically sound, and do the data support the conclusions?

Reviewer #1: Yes

Reviewer #2: Yes

Reviewer #3: Yes

2. Has the statistical analysis been performed appropriately and rigorously? 

Reviewer #1: Yes

Reviewer #2: Yes

Reviewer #3: Yes

3. Have the authors made all data underlying the findings in their manuscript fully available?

Reviewer #1: Yes

Reviewer #2: Yes

Reviewer #3: Yes

4. Is the manuscript presented in an intelligible fashion and written in standard English?

Reviewer #1: Yes

Reviewer #2: Yes

Reviewer #3: Yes

5. Review Comments to the Author

Reviewer #1: The manuscript mainly presents an analysis study of OpenSCAD and compare it with FreeCAD Python scripts. The authors also created a parametric open source hardware design to compare these tools.

The manuscript is of interest to the PlosOne readership. The introduction provides sufficient background and include all relevant references. The research design is appropriate. The methods adequately described. The results clearly presented. The scientific style of presentation is high enough.

However, the introduction is too long, I would recommend authors to summaries the motivation.

(Line 73) The authors mentioned some other FOSS CAD tools available to create solid models using a programming language, such as FreeCAD, PythonOCC, CadQuery and BRL-CAD. Among these CAD tools, Among these CAD tools, we have found FreeCAD to be the most active, having around a major release each year,..etc. Could you please compare the interesting features of OpenSCAD and FreeCAD with FreeCAD, PythonOCC, CadQuery and BRL-CAD.

(line 262) *.scad

The main references are WWW, we recommend the authors to cite research articles if applicable. The readership usually couldn’t reach web sites for some cases and that happens for many reasons for example changing/removing the web address when a company acquired by another compony.

The flowing references are a web address; 2, 3, 4, 7, 12, 14, 21,22,30, 34, 3,36,37,38,39,40, 43, 44, 45, 46, 47, 48, 49, 50, 51, 52, 53, 54, 56, 57, and 58.

Reference [5] missing the doi.

Reference [9] missing the doi.

Reference [13] missing the doi.

Reference [15] is a book but missing the doi.

Reference [16] please check the order of the authors; Luis Felipe Rosado Murillo then Matti Veikko Pietari Kauttu.

Reference [32] missing doi. doi.org/10.1007/978-3-662-44468-9_53.

Reference [42] missing doi. 10.1201/9781315119601.

Reviewer #2: After reviewing the manuscript entitled “Parametric CAD modeling for open source scientific hardware: comparing OpenSCAD and FreeCAD Python scripts” (PONE-D-19-22759).

Here are my reviewing points:

The manuscript comparing OpenSCAD and FreeCAD. The work did provide some insight that can potentially guide the developers to improve the CAD modeling using open source scientific hardware.

1. The comparison was comprehensive and the authors provided sufficient resources.

2. The authors provided sufficient and good literature background.

3. The featured of both OpenSCAD and FreeCAD were analyzed fairly.

4. The major drawback is that this manuscript fits to be a case study rather than a research article. However, in both cases it is still valuable and interesting.

5. typos line (73) (The) should be (There).

6. In table (7), the authors mentioned that it is divided into (three) topics, but it seems that there are (four) topics discussed in the table as mentioned in line (238), which are (the geometric modeling kernel, the usability, the programming language characteristics and the tool features), so its need to be corrected to (four) topics in the title of the table.

7. The authors have to follow one format for the references, and I would suggest them to mentioned the access time for websites.

Reviewer #3: I know several groups working on parametric CAD modeling for open source scientific hardware and they demonstrated progress however there is no single comprehensive study compares such type of CADs (in my knowledge).

The authors present an analysis study of prevalent modeling CADs (OpenSCAD and FreeCAD). They created a parametric open source hardware design to compare OpenSCAD and FreeCAD . The authers also describe methodology of comparison and they provided necessary scripts and supplementary documents.

The main advantage of this paper is the systematic analysis. It feels that the authors predicted readership’s questions, so they put consequent answers. The paper is of interest for many PlosOne’s auditory. The done works of other groups and researchers are accurately acknowledged, I would recommend to publish this manuscript after minor corrections.

My main concern is about the motivation for the authors to publish their study in a journal while they have already deposited scripts elsewhere (github). I would recommend authors to explain that in the introduction. The bibliography needs some attention according to the journal instructions.

6. PLOS authors have the option to publish the peer review history of their article (what does this mean?). If published, this will include your full peer review and any attached files.

Reviewer #1: No

Reviewer #2: No

Reviewer #3: No

---

## [Author Response · Author response to Decision Letter 0]

28 Oct 2019

Reviewer #1:

However, the introduction is too long, I would recommend authors to summaries the motivation.

- We have attempted to reduce the introduction length, but since other reviewer believed that the introduction was adequate, we were afraid to remove important background information. Thus, we have barely been able to reduce it.

---

(Line 73) The authors mentioned some other FOSS CAD tools available to create solid models using a programming language, such as FreeCAD, PythonOCC, CadQuery and BRL-CAD. Among these CAD tools, Among these CAD tools, we have found FreeCAD to be the most active, having around a major release each year,..etc. Could you please compare the interesting features of OpenSCAD and FreeCAD with FreeCAD, PythonOCC, CadQuery and BRL-CAD.

- The reviewer's comment is relevant. We have created a new section with an extended explanation of the features of these FOSS CAD tools. In this section we have explained why we have chosen FreeCAD and also have included the FreeCAD CadQuery workbench in the subsequent analysis. This FreeCAD workbench facilitates the use of Python scripts in FreeCAD, and thus, supports the use of FreeCAD. We believe that with this addition the manuscript is more comprehensive. Besides, we have added included the CadQuery models in the software repository.

---

(line 262) *.scad

- Corrected.

---

The main references are WWW, we recommend the authors to cite research articles if applicable. The readership usually couldn’t reach web sites for some cases and that happens for many reasons for example changing/removing the web address when a company acquired by another compony.

The flowing references are a web address; 2, 3, 4, 7, 12, 14, 21,22,30, 34, 3,36,37,38,39,40, 43, 44, 45, 46, 47, 48, 49, 50, 51, 52, 53, 54, 56, 57, and 58.

- The WWW are mainly from computer programs, associations and projects statistics that we have not been able to find research articles with that information. However, all the WWW references have been archived and should be available at any time with the information at the time of archive.

---

Reference [5] missing the doi.

- Corrected. In the revised manuscript, this reference is [4].

Reference [9] missing the doi.

- Corrected. In the revised manuscript, this reference is [8]. 

Reference [13] missing the doi.

- We could not find the doi, but we added where it is available. In the revised manuscript, this reference is [12]. 

Reference [15] is a book but missing the doi.

- We could not find the doi, but the ISBN (9780133373905) could be added if necessary. In the revised manuscript, this reference is [14].

Reference [16] please check the order of the authors; Luis Felipe Rosado Murillo then Matti Veikko Pietari Kauttu.

- Corrected. In the revised manuscript, this reference is [15].

Reference [32] missing doi. doi.org/10.1007/978-3-662-44468-9_53.

- Corrected.

Reference [42] missing doi. 10.1201/9781315119601.

- Corrected. In the revised manuscript, this reference is [53].

Reviewer #2:

5. typos line (73) (The) should be (There).

- Corrected.

6. In table (7), the authors mentioned that it is divided into (three) topics, but it seems that there are (four) topics discussed in the table as mentioned in line (238), which are (the geometric modeling kernel, the usability, the programming language characteristics and the tool features), so its need to be corrected to (four) topics in the title of the table.

- Corrected.

7. The authors have to follow one format for the references, and I would suggest them to mentioned the access time for websites.

- The format of the references have been corrected. We have added the archived link with its archive date in order to have a permanent link at the time of archive.

-----------------

Reviewer #3:

My main concern is about the motivation for the authors to publish their study in a journal while they have already deposited scripts elsewhere (github). I would recommend authors to explain that in the introduction. 

- We have deposited the scripts in github to be in compliance with PLOS ONE software sharing policy: “We expect that all researchers submitting to PLOS submissions in which software is the central part of the manuscript will make all relevant software available without restrictions upon publication of the work”.

In the third section (Open hardware models created as test-bench), in line 172 of the revised marked-up copy, we have included an explanation about the availability of the associated scripted models.

---

The bibliography needs some attention according to the journal instructions.

- Corrected.

---

## [Decision Letter · Decision Letter 1]

13 Nov 2019

Parametric CAD modeling for open source scientific hardware: comparing OpenSCAD and FreeCAD Python scripts

PONE-D-19-22759R1

Dear Dr. Machado,

We are pleased to inform you that your manuscript has been judged scientifically suitable for publication and will be formally accepted for publication once it complies with all outstanding technical requirements.

With kind regards,

Talib Al-Ameri, Ph.D

Academic Editor

PLOS ONE

Additional Editor Comments (optional):

Reviewers' comments:

Reviewer's Responses to Questions

**Comments to the Author**

1. Reviewer 3 has repeated the same comments as in round 1. I think the authors have adequately addressed all comments raised in a previous round of review and I feel that this manuscript is now acceptable for publication.

Reviewer #1: (No Response)

Reviewer #2: All comments have been addressed

Reviewer #3: All comments have been addressed

2. Is the manuscript technically sound, and do the data support the conclusions?

Reviewer #1: (No Response)

Reviewer #2: Yes

Reviewer #3: Yes

3. Has the statistical analysis been performed appropriately and rigorously? 

Reviewer #1: (No Response)

Reviewer #2: Yes

Reviewer #3: Yes

4. Have the authors made all data underlying the findings in their manuscript fully available?

Reviewer #1: (No Response)

Reviewer #2: Yes

Reviewer #3: Yes

5. Is the manuscript presented in an intelligible fashion and written in standard English?

Reviewer #1: (No Response)

Reviewer #2: Yes

Reviewer #3: Yes

6. Review Comments to the Author

Reviewer #1: (No Response)

Reviewer #2: The authors have addressed my comments that I had raised in a previous round of review and I feel that this manuscript is now acceptable for publication.

The paper is well organized and clearly written. The technical quality of the manuscript is satisfactory. The length of the manuscript is now seems good, where most repeated sections are manipulated.

Reviewer #3: (same comments of round 1)

7. PLOS authors have the option to publish the peer review history of their article (what does this mean?). If published, this will include your full peer review and any attached files.

Reviewer #1: No

Reviewer #2: No

Reviewer #3: Yes

---

## [Editor Report · Acceptance letter]

19 Nov 2019

PONE-D-19-22759R1 

Parametric CAD modeling for open source scientific hardware: comparing OpenSCAD and FreeCAD Python scripts 

Dear Dr. Machado:

I am pleased to inform you that your manuscript has been deemed suitable for publication in PLOS ONE. Congratulations! Your manuscript is now with our production department. 

With kind regards,

on behalf of

Dr. Talib Al-Ameri 

Academic Editor

PLOS ONE